# Gestational Factors throughout Fetal Neurodevelopment: The Serotonin Link

**DOI:** 10.3390/ijms21165850

**Published:** 2020-08-14

**Authors:** Sabrina I. Hanswijk, Marcia Spoelder, Ling Shan, Michel M. M. Verheij, Otto G. Muilwijk, Weizhuo Li, Chunqing Liu, Sharon M. Kolk, Judith R. Homberg

**Affiliations:** 1Department of Cognitive Neuroscience, Donders Institute for Brain, Cognition and Behavior, Radboud University Nijmegen Medical Centre, 6525 EN Nijmegen, The Netherlands; sabrina.hanswijk@radboudumc.nl (S.I.H.); Marcia.Spoelder-Merkens@radboudumc.nl (M.S.); Michel.Verheij@radboudumc.nl (M.M.M.V.); Otto.Muilwijk@radboudumc.nl (O.G.M.); 2Netherlands Institute for Neuroscience, an Institute of the Royal Netherlands Academy of Arts and Sciences, 1105 BA Amsterdam, The Netherlands; l.shan@nin.knaw.nl; 3College of Medical Laboratory, Dalian Medical University, Dalian 116044, China; liweizhuoya@hotmail.com (W.L.); liuchunqing@dmu.cn (C.L.); 4Department of Molecular Neurobiology, Donders Institute for Brain, Cognition and Behavior, Radboud University, 6525 AJ Nijmegen, The Netherlands; s.kolk@donders.ru.nl

**Keywords:** gestation, serotonin, neurodevelopment, neural circuit formation, neuropsychiatric disorders

## Abstract

Serotonin (5-HT) is a critical player in brain development and neuropsychiatric disorders. Fetal 5-HT levels can be influenced by several gestational factors, such as maternal genotype, diet, stress, medication, and immune activation. In this review, addressing both human and animal studies, we discuss how these gestational factors affect placental and fetal brain 5-HT levels, leading to changes in brain structure and function and behavior. We conclude that gestational factors are able to interact and thereby amplify or counteract each other’s impact on the fetal 5-HT-ergic system. We, therefore, argue that beyond the understanding of how single gestational factors affect 5-HT-ergic brain development and behavior in offspring, it is critical to elucidate the consequences of interacting factors. Moreover, we describe how each gestational factor is able to alter the 5-HT-ergic influence on the thalamocortical- and prefrontal-limbic circuitry and the hypothalamo-pituitary-adrenocortical-axis. These alterations have been associated with risks to develop attention deficit hyperactivity disorder, autism spectrum disorders, depression, and/or anxiety. Consequently, the manipulation of gestational factors may be used to combat pregnancy-related risks for neuropsychiatric disorders.

## 1. Introduction

The neurotransmitter serotonin (5-HT) plays a major role in neuropsychiatric disorders and is a key player in brain development [1,2]. In adulthood, 5-HT cannot cross the blood–brain barrier, while its precursors tryptophan and 5-hydroxytryptophan (5-HTP) can (see review [3]). However, roughly during the first half of pregnancy, 5-HT can be transferred from the mother’s placenta, via the fetal periphery, to the fetal brain [4,5,6]. It is, therefore, likely that the maternal environment can greatly influence fetal brain development. This, in turn, may play a role in the onset of multiple neuropsychiatric disorders, as many of these disorders seem to have a developmental origin [7]. Further insight into how gestational factors can influence the fetal 5-HT system may advance our understanding of the aetiology underlying several neuropsychiatric disorders.

### 1.1. The 5-HT System and Its Embryonic Development

As illustrated in Figure 1, 5-HT signaling is tightly controlled by (1) tryptophan hydroxylase (TPH), the rate-limiting enzyme of 5-HT synthesis, and by (2) the 5-HT transporter (5-HTT), which is responsible for the high-affinity reuptake of 5-HT. Two TPH isoforms have been identified to synthesize 5-HT from tryptophan. TPH1 is responsible for 5-HT synthesis in the periphery and to a smaller extent also in the brain pineal gland. TPH2 is essential for 5-HT synthesis in the brain [8,9]. In addition, TPH2 is also important for 5-HT synthesis in the enteric nervous system and thereby the regulation of gastrointestinal motility [10]. In the brain, 5-HT is synthesized by 5-HT-ergic neurons located in the various nuclei of the raphe nuclei in the brainstem and is transported into synaptic vesicles by the vesicular monoamine transporter isoform 2. The exocytosis of these vesicles releases 5-HT into the synaptic cleft, where it binds to post-synaptic 5-HT receptors. Thus far, at least 14 different 5-HT receptor subtypes have been identified (5-HT_1A/B/D/E/F_, 5-HT_2A/B/C_, 5-HT_3_, 5-HT_4_, 5-HT_5A/B_, 5-HT_6_, and 5-HT_7_), which can trigger a variety of signaling cascades. All but the 5-HT_3_ receptor are metobotropic and belong to the G protein-coupled receptor superfamily. A complete overview of the receptor’s transduction mechanisms is beyond the scope of this review and has been discussed in detail previously [11,12]. Of note, at the cost of 5-HT, tryptophan can also serve as a precursor of kynurenine via the rate-limiting enzymes tryptophan 2,3-dioxygenase (TDO) and indoleamine 2,3-dioxygenase (IDO) [13]. Kynurenine is further degraded into multiple neuroactive metabolites, including kynurenic acid and quinolinic acid. The kynurenine pathway plays a role in immune response and inflammation [14,15]. 

The various developmental stages of the fetal 5-HT system are largely comparable between humans and rodents. In humans, 5-HT-ergic neurons are first evident in the hindbrain at gestational week 5 [16] after which they project to the forebrain, reaching the cerebral cortex around gestational week 10 and the most rostral telencephalon around gestational week 13 [17]. At gestational week 15, during the second trimester of human pregnancy, the typical organization of clustered 5-HT cell bodies into the raphe nuclei is observed [18]. Similarly, in rodents, 5-HT-positive cells are observed in the raphe nuclei around embryonic day (E) 10–12, which corresponds to the second trimester of pregnancy in humans. One day after their appearance, these cells are able to synthesize 5-HT [2]. 5-HT-ergic axons reach the forebrain in mice around E16.5 and in rats around E17.5 [19,20]. Interestingly, 5-HT-ergic signaling molecules are already present in the forebrain before 5-HT-ergic axons reach it. Human data show the expression of 5-HTT in the brain around gestational week 8, i.e., the first trimester of pregnancy [21]. Similarly, in mice and rats, brain 5-HTT is detected around E8-9 in the forebrain [2,22,23,24]. This suggests that an exogenous source may provide the fetal forebrain with 5-HT before the actual 5-HT-ergic projections reach the forebrain.

### 1.2. Maternal and Placental Sources of 5-HT

Two exogenous sources have been reported to supply the fetus with 5-HT: the placenta and maternal platelets, i.e., 5-HT overflow from the gut. Although these two sources are clearly associated with the fetal 5-HT supply, it is still quite unclear what the varying degrees of supply from the placenta and maternal platelets are and whether they act in parallel [25]. Two studies reported that the placenta expresses both maternal and fetal isoforms of TPH1 and TPH2 (mouse placenta [19] and human placenta [26]), whereas another study was not able to detect placental TPH1 (mouse and human placentas [25]). It is, therefore, not fully clear as to whether the placenta has the necessary machinery to synthesize 5-HT from maternal tryptophan. Other literature suggests that maternal platelets serve as the exogenous source for the delivery of 5-HT to the fetus [5,25]. Maternal platelets degranulate in the intervillous space of the placenta, thereby releasing maternal 5-HT into the placenta. Subsequently, this maternal 5-HT is transferred through the placenta to the fetus. This transfer is facilitated by, among others, the fetal isoform of 5-HTT and organic cation transporter 3 (OCT3) [25]. However, Bonnin and colleagues could not find support for the hypothesis that maternal platelets serve as the exogenous source of 5-HT to the fetus. That is, 5-HT levels in the forebrain of 5-HTT heterozygotic embryos from 5-HTT knockout dams did not differ from those of 5-HTT heterozygotic embryos from 5-HTT wildtype dams [19]. It should be mentioned that the placenta does express monoamine oxidase A (MAOA), an enzyme responsible for 5-HT breakdown [27]. Hence, this enzyme may influence placental 5-HT levels and thus control the regulation of fetal 5-HT. Despite the unclarities, the placenta-derived 5-HT content (referring to both placenta and platelet 5-HT supply) is expected to play vital roles in the regulation of fetal processes. This exogeneous source is mainly provided to the fetus throughout the first half of the pregnancy, which is around the same time as 5-HT-regulated developmental events including corticogenesis and circuitry maturation occur [2]. For this reason, gestational factors affecting not only an offspring’s raphe-produced 5-HT content but also placenta-derived 5-HT content can have substantial effects on an offspring’s early brain development. 

### 1.3. Aim of Review

We performed an extensive literature search up to February 2020 using the PubMed database. The key characteristics of referred human and animal studies are shown in Appendix A, respectively. The objective of this review is to carry out a comprehensive approach towards gestational factors affecting the fetal 5-HT system and as a result may affect an offspring’s brain development and behavior regarding neuropsychiatric disorders. We first introduce this topic by providing background information concerning 5-HT-ergic neuromodulatory effects on brain development (Section 2). Thereafter, we highlight what is currently known about the effects of gestational factors on an offspring’s brain 5-HT system and neuropsychiatric disorders related to brain development and behavior (Section 3). Gestational factors include maternal 5-HT-ergic genotype, 5-HT-related diet composition, stress, 5-HT-related medication, and immune activation. Finally, we discuss the (interactive) impact of the gestational factors on the placenta-derived and offspring’s raphe-produced 5-HT content, the related brain circuit development, and how these effects may provide risk of the onset of specific neuropsychiatric phenotypes later in life (Section 4). Noteworthy paternal and postnatal environmental factors, such as paternal genotype and maternal care, may influence an offspring’s 5-HT system and development as well. These factors have been discussed in detail previously (e.g., [28]) and are beyond the scope of this review.

## 2. Modulatory Effects of 5-HT on Brain Development

During development, 5-HT in the brain is involved in a variety of neurodevelopmental processes, such as neuronal proliferation and migration, as well as initial axon targeting and circuitry maturation [2]. Of note, 5-HTT and the vesicular monoamine transporter are transiently expressed on forebrain glutamatergic neurons [2,29]. This expression is observed before raphe nuclei-derived 5-HT-ergic projections are born until the second week of postnatal life [29]. Glutamatergic neurons in the forebrain are located in various cortical structures that play a role in complex behavior including the sensory cortex, prefrontal cortex (PFC), and hippocampus [2,30]. Furthermore, glutamatergic neurons project from the thalamus to the cortex (i.e., thalamocortical afferents). 5-HT accumulation in these glutamatergic neurons modulates responsiveness of neurons to guidance cues, and 5-HT may function itself as a guidance cue as well [31,32].

5-HT receptors within the brain are expressed on neurons as well as glial cells. As shown in Table 1, activation of neuronal 5-HT receptors is involved in the above-mentioned neurodevelopmental events (for more details see Section 2.1 and Section 2.2). Activation of astrocytic 5-HT receptors have been reported to play a role in neurogenesis and the release of the neurite-extending growth factor S100β (see review [11]). Interestingly, 5-HT regulates astrocytic activity, a cell type that has an important role in modulating synaptic function and network activity within a variety of brain areas including the PFC [33]. The microglial-expressed 5-HT_2B_ receptor regulates 5-HT neurotransmission by releasing nitric oxide, cytokines, and glutamate, which, in turn, affect 5-HTT-mediated 5-HT clearance [34]. Moreover, microglia play an important role in brain maturation and inflammation. They are involved in cell positioning and survival and in synaptic patterning within various cortical structures including the sensory cortex, PFC, and hippocampus [35]. Regarding oligodendrocytes, elevated extracellular 5-HT levels negatively affect 5-HT_2A_-dependent oligodendrocytes’ survival (i.e., aberrant outgrowth, increased development-dependent cell death) leading to reduced myelination [36]. The recent investigations regarding the function of 5-HT and 5-HT receptors in the different glial cells are exciting and might result in refreshing new insights into the role of 5-HT in the development of these brain cells. Thus far, most studies have investigated the role of 5-HT on neuronal development.

### 2.1. Corticogenesis and Circuit Maturation

During embryonic development, 5-HT projections to the forebrain regulate certain aspects of corticogenesis, including neuronal migration and the proper positioning of pyramidal neurons and interneurons [2,37,38,39]. It has been shown that 5-HT-ergic fibers within cortical regions, including the medial PFC (mPFC), contact Cajal Retzius cells, and thereby possibly affect reelin release [40,41,42]. In turn, reelin affects the correct placement of pyramidal neurons in cortical layers [43]. How exactly 5-HT affects migration of pyramidal cells is still unclear. Another course of action via which the 5-HT system can influence neuronal migration involves the 5-HT receptors. The 5-HT_6_ receptor has been shown to mediate pyramidal neuron migration via controlling the activity of CDK5/p35, a master regulator of pyramidal neuron migration [44]. Interneuron migrational events are mediated through both the 5-HT_6_ and 5-HT_3_ receptors [39,44,45,46,47]. More specifically, activation of the 5-HT_3_ receptor increases calcium currents into the cell, thereby promoting the migratory speed of cortical interneurons during the invasion of the cortical plate. The modulation of calcium transients is speculated to be through the regulation of N-methyl-D-aspartate receptors, also known as the NMDA receptors, and voltage-gated calcium channels [45,46].

5-HT also affects several other aspects of circuitry maturation, such as neurite outgrowth, pruning, and the remodeling of dendritic spines [48,49,50,51]. For example, activation of 5-HT receptors enhances neurite outgrowth and leads to neuronal survival [52,53]. The influence of 5-HT in neurite outgrowth is furthermore studied using a Pet-1 knockout animal model. Pet-1 is a key transcriptional regulator that controls 5-HT neuronal levels: depletion of this regulator results in a 70% reduction of raphe 5-HT neurons and an 80% reduction of forebrain 5-HT levels due to a diminished cortical innervation by 5-HT projections [54]. Such diminished 5-HT-ergic innervation leads to alterations in cortical interneuron activity as well as cortical neuron identity and placement and may result in an increase in cortical network excitability [55]. Pet-1 knockout mice also show elevated sensitivity to 5-HT_1_ and 5-HT_2_ receptors, resulting in spontaneous synaptic activity [56]. 

### 2.2. Brain Circuits’ Structure and Function

Due to 5-HT’s role in neurodevelopmental events, changes in 5-HT levels can impact the maturation and the functioning of a variety of brain circuits, including thalamocortical and prefrontal-limbic circuits, as well as the hypothalamo-pituitary-adrenocortical (HPA)-axis. As described below, these circuits play an important role in the aetiology of neuropsychiatric disorders.

The thalamocortical projections mediate somatosensory (tactile) perceptions. Areas involved are the thalamus, the primary somatosensory barrel cortex, and the secondary somatosensory cortex. 5-HT accumulation in thalamocortical afferents plays a critical role in the formation of sensory cortices. Primary somatosensory barrel cortex development is greatly dependent on extracellular 5-HT concentrations regulated by the transient 5-HTT and 5-HT_1B_ receptor expression on these thalamocortical afferents [32,57]. Activation of the 5-HT_1B_ receptor has been shown to negatively regulate presynaptic glutamate release [58]. In turn, this negative regulation can instruct the clustering of postsynaptic excitatory synapses in the typical barrel formation [59]. Presynaptic 5-HT_2A_ receptors, which are located at thalamocortical synapses throughout life, additionally control the NMDA-operated induction of thalamofrontal connectivity and the associated cognitive functions [60]. Moreover, somatosensory cortical inhibition, and, in turn, sensory processing, is affected by 5-HT-regulated astrocytic purinergic signaling [61]. Alterations in 5-HT levels can thus lead to structural and functional reorganization of thalamocortical afferents as well as intracortical microcircuitry. Altered development of sensory cortices will result in changes in the perception of sensory stimuli early in life. It has been shown that alterations in this circuitry can contribute to the development of pathologies, such as autism spectrum disorders (ASD), attention deficit hyperactivity disorder (ADHD), and depression [62,63].

Another circuitry influenced by alterations in 5-HT levels is the prefrontal-limbic circuitry. This circuitry plays a crucial role in stress and emotional responses [64]. Areas include the mPFC, basal ganglia, amygdala, and hippocampus. 5-HT plays a prominent role in this circuitry as multiple receptors (e.g., 5-HT_1A_ and 5-HT_2A_) are expressed in these areas [65]. In the mPFC, these receptors are mainly involved in modulating cortical activity. Whereas 5-HT_1A_ receptors inhibit pyramidal neurons and interneurons via activation of GIRK channels, the 5-HT_2A_ receptors excite these neurons via unknown mechanisms [66]. Indeed, drug-induced alterations in 5-HT_2A_ receptor activity influence attention and impulsive behavior (see review [66]). This circuitry is often involved in the onset of various neuropsychiatric disorders, such as ASD, ADHD, anxiety, and depression [67,68].

The HPA-axis is yet another circuitry whose development can be influenced by alterations in 5-HT levels. It is a crucial neuroendocrine signaling system involved in physiological homeostasis and the adrenal glucocorticoid stress response [69]. The HPA-axis involves the hypothalamus where corticotrophin releasing factor is being produced, the pituitary that responds to corticotrophin releasing factor to produce the hormone adrenocorticotropin, and the adrenal cortex that responds to adrenocorticotropin to produce cortisol (in humans) or corticosterone (in rodents). Activation of this axis occurs via a discrete set of neurons in the hypothalamic paraventricular nucleus. These neurons are influenced by indirect input from limbic system-associated regions, including the mPFC, hippocampus, and amygdala. Accumulating data suggest a regulation of the corticotrophin releasing factor neuronal activity and thus activation of the HPA-axis via 5-HT_1A_ and 5-HT_2A_ receptors at the hypothalamus and via 5-HT_2C_ receptors at the hypothalamic paraventricular nucleus [70,71]. A change in 5-HT levels results in altered 5-HT-ergic innervations in the hypothalamic paraventricular nucleus and the hippocampus [72] and thereby can severely affect basal HPA-axis activity, predominantly at the level of the adrenal gland [73]. This, in turn, can lead to severe hormonal imbalances. Dysregulation of the HPA-axis is strongly implicated in the pathology of mood disorders, such as depression and anxiety disorders [74,75,76].

## 3. Gestational Factors Influence Fetal Development during the Embryonic Period and after Birth

### 3.1. 5-HT-ergic Maternal Genotype Influences the 5-HT System, Neurodevelopment, and Behavior in the Offspring

As proposed previously by Gleason and colleagues, maternal genetic variations affecting the 5-HT system (hereafter called maternal 5-HT-ergic genotype) are able to influence fetal development [77]. Changes in the 5-HT system can occur through alterations in placental and fetal 5-HT levels or through transmission of risk alleles. The effects of the transmission of risk alleles on an offspring’s brain development and behavior are discussed in other reviews (e.g., [21]). In short, maternal 5-HT-ergic genotype perturbs fetal 5-HT homeostasis by increasing or decreasing 5-HT supply. The animal studies addressed below suggest that the thalamocortical circuitry is particularly sensitive to maternal variance in 5-HT-ergic genotype. 

#### 3.1.1. Associations with Behavioral Changes in Offspring in Humans

A variety of human studies link maternal 5-HT-ergic genotype to differences in an offspring’s brain structure and function, as well as behavior. A well-known and common polymorphic variant is the 5-HT transporter-linked polymorphic region (5-HTTLPR). This polymorphism consists of two variants, the short (S)-allelic and long (L)-allelic variant. The S-allelic variant confers reduced 5-HTT gene transcription, thereby reducing 5-HTT’s availability in cultured human lymphoblast cells and in the brain as measured via single photon emission computed tomography [78,79]. In one study, children heterozygous for 5-HTTLPR were divided into two groups depending on maternal 5-HTTLPR genotype, thus either S/S or L/L. Children from mothers homozygotic for the S allele were found to lift their pencils less often in a visuomotor performance task in comparison to children from mothers homozygotic for the L allele. This finding correlated with changes in their somatosensory cortex grey matter density. That is, children with denser grey matter lifted their pencils less often [80]. Interestingly, regarding associated neuropsychiatric disorders, children of 5-HTTLPR L-allele carrying mothers have a higher risk of ASD [81], while no association was found between 5-HTTLPR allelic variants and the occurrence of depression in mother–child groups [82]. Lower levels of maternal whole-blood 5-HT levels measured at mid-pregnancy were associated with the most severely affected ASD phenotypes [83]. The offspring of mothers carrying TPH1 loss-of-function mutations exhibited 1.5- to 2.5-times higher ADHD scores in comparison to controls and the offspring from fathers carrying such mutations [84]. Future studies with whole genome sequencing between mother and child and the neurobehavioral development of the child could potentially yield a guidance to whether which genotype combination exactly has an increased risk of the development of neuropsychiatric disorders.

#### 3.1.2. 5-HT System Alterations in the Animal Offspring

To follow up on the human data, a variety of animal studies have linked maternal 5-HT-ergic genotype to structural and functional alterations in the offspring. For example, studies reported that maternal 5-HT_1A_ receptor deficiency is associated with reduced juvenile ultrasonic vocalizations, adult anxiety-like traits in the offspring, increased stress responsiveness, and delayed development of the ventral dentate gyrus, independent of an offspring’s genotype [85,86]. Furthermore, mouse fetuses, capable of producing their own 5-HT but conceived by a TPH1 knockout mother (i.e., lacking 5-HT in the periphery), showed a reduced mitotic activity within the roof of the neocortex (future cerebral cortex) together with dramatic abnormalities in the shape of this region and of hindbrain regions [5]. In support, E18 TPH1 knockout embryos from TPH1 knockout dams showed also a decrease in proliferation rate in comparison to TPH1 wildtype embryos from TPH1 wildtype dams [87]. Another way to investigate maternal 5-HT-ergic genotype effects is the use of the 5-HTT Ala56 gain-of-function mutation mouse model [88]. In general, 5-HTT Ala56 mice display increased CNS 5-HT clearance, enhanced 5-HT receptor sensitivity, elevated peripheral blood 5-HT levels, and increased ASD-like behavior [89]. Offspring from Ala56 mothers displayed decreased placental and embryonic forebrain 5-HT levels and a broadening of 5-HT-sensitive somatosensory thalamocortical afferents at E14.5. The forebrain 5-HT homeostasis recovered at E18.5 [6], when 5-HT raphe fibers had reached the forebrain. 

### 3.2. Maternal 5-HT-ergic-Related Diets Influence the Tryptophan Pathway, Neurodevelopment, and Behavior in the Offspring

Another route via which the mother can influence her offspring’s development involves her diet. Nutrients received from the mother via the placenta or during lactation are crucial for an offspring’s health and growth [90,91]. In the following sections, we describe the effects of maternal tryptophan-related diets, high-fat diets, and alcohol consumption on 5-HT-ergic brain development of the offspring. In short, depending on the composition of the diet, the mother can perturb fetal 5-HT homeostasis by increasing or decreasing fetal 5-HT availability and influencing the fetal 5-HT system in the hippocampus, hypothalamus, amygdala, the raphe nuclei, and the mPFC.

#### 3.2.1. Maternal Tryptophan-Related Diets Affect the Placental Tryptophan Pathway in Animals

The most obvious nutritional component that can affect 5-HT-ergic development in the offspring is tryptophan, the essential amino acid that serves as the precursor of 5-HT. Tryptophan is amongst others found in salmon, poultry, eggs, spinach, seeds, milk, soy products, and nuts. The greatest part of the consumed tryptophan is directly metabolized in the liver, the rest binds to albumin and circulates freely throughout the body [92]. Tryptophan supplements can be consumed to increase tryptophan availability and enhance 5-HT synthesis. Already in 1991, it was shown that oral administration of tryptophan during gestation in rodents increased fetal whole brain tryptophan and 5-HT levels [93]. A diet enriched with tryptophan (i.e., 50 g/kg) resulted in increased maternal, placental, and fetal tryptophan levels. As a result, both placental development and fetal growth were reduced, and pup death rate increased [94]. Less enriched tryptophan diets (i.e., ±10 g/kg) also resulted in peripheral hyperserotonemia and increased TPH activity in the gastrointestinal tract of dams and their pups [95,96,97]. However, opposed effects in the offspring’s brain (in contrast to the gastrointestinal tract) were reported due to low tryptophan enriched diets (i.e., ±10 g/kg). Namely, decreased frontal cortex 5-HT levels and TPH activity were found, probably caused by a delay in 5-HT axon outgrowth [96]. Remarkably, chronic deprivation of tryptophan (in contrast to enriched tryptophan diets) also resulted in decrements of dorsal raphe 5-HT-ergic neuron populations and neuronal migration disturbances, leading to an altered topography of the raphe nucleus (i.e., rostralization instead of caudalization of the dorsal raphe nucleus) [98]. Consequently, tryptophan-deprived diets resulted in decreased 5-HT levels in the PFC and decreased hippocampal but increased striatal 5-HT metabolism in adult offspring [99]. Such a diet deprived of tryptophan also resulted in a decrease in striatal brain derived neurotrofic factor (BDNF) levels and a reduction in hippocampal proliferation rate and in neuronal outgrowth, i.e., reduced dendritic spine density and abnormal dendrite swelling [99,100]. Notably, active maternal care seems to be reduced by a tryptophan-deprived diet and thus may aggravate the above described effects [99]. Depressive-like behavior in the offspring has also been reported upon neonatal tryptophan deprivation [99], which may be due to either a reduction in maternal care, disturbed brain maturation, or their interaction. 

To ensure the transport of tryptophan from the mother to the fetus, a diet high in carbohydrates and low in proteins has been suggested [101]. Carbohydrates evoke insulin secretion. Insulin, in turn, reduces most large amino acids, with the exception of tryptophan [102]. In humans and rodents, the transfer of placental tryptophan from maternal blood into the fetal circulation is predominantly mediated by the so-called system A and system L transporters [103,104]. System A functions in a sodium-dependent manner to mediate the uptake of nonessential amino acids into the cell. In this way, a gradient is created that is used to drive the exchange for extracellular essential amino acids, such as tryptophan, via sodium-dependent system L [105]. Excessive amounts of nonessential amino acids in the maternal blood thus might increase the maternal-fetal transfer of essential amino acids, including tryptophan. Multiple animal studies investigated the effects of gestational low-protein diets on an offspring’s brain 5-HT system. Notably, some studies used isocaloric protein diets in which the low protein diets were adjusted with a higher content of carbohydrates, thereby creating low-protein high-carbohydrate diets. Such low-protein high-carbohydrate diets reduced maternal plasma tryptophan levels and increased maternal plasma 5-HT levels and the offspring’s brain 5-HT levels after birth [106,107]. It remains unclear whether the exposure to such diets decreases or increases placental and fetal 5-HT levels [106,107]. When using a low-protein high-carbohydrate diet supplemented with tryptophan, neither the maternal nor fetal 5-HT system is altered [108]. Importantly, when exposed to a prenatal carbohydrate-restricted diet consisting of similar protein levels, and thus tryptophan levels, as the control diet, fetal whole brain concentrations of tryptophan, 5-HT, 5-HIAA were reduced [109]. 

With regard to lasting effects within specific brain regions in the offspring, (isocaloric) maternal low-protein diets, irrespective of the gestational period, resulted in increased 5-HT levels in the mPFC, hypothalamus, and hippocampus throughout life [106,110,111,112]. However, a few studies were not able to replicate adult hypothalamic and hippocampal 5-HT increments [107,113,114,115]. Not only offspring’s 5-HT levels but also 5-HT-ergic receptors are influenced due to gestational exposure to low-protein diets. A prenatal isocaloric low-protein diet resulted in a reduced expression of the hypothalamic 5-HT_2C_ receptor at birth and throughout life [107]. Similarly, a reduced hippocampal 5-HT_1A_ receptor functionality has been reported [113,116]. Interestingly, Ye and colleagues recently reported that the reduced 5-HT_1A_ receptor functionality was accompanied by an increased sensitivity to stress throughout life [116].

#### 3.2.2. Maternal High-Fat Diet Reduces 5-HT Production in the Animal Offspring 

Maternal high-fat diets throughout pregnancy result in an increased susceptibility to developing ASD, ADHD, anxiety, and depression. Potential mechanisms underlying offspring behavioral changes include maternal increases in nutrients (glucose and fatty acids) and hormones (insulin and leptin), maternal and placental increases in oxidative stress, and inflammatory cytokines. In turn, these maternal dysregulations increase the risk of placental dysfunction, as well as an increased inflammation in the fetal brain, alterations in the dopaminergic, and 5-HT-ergic systems in the fetal brain and perturbations in synaptic plasticity and the HPA-axis (see reviews [117,118,119,120]). Indeed, a perinatal high-fat diet in primates revealed an increase in fetal dorsal raphe nucleus TPH2 gene expression, and after birth, decrements in 5-HT levels in the cerebrospinal fluid, a reduction in TPH2 mRNA expression in the dorsal and median raphe nuclei, and an upregulation of the 5-HT_1A_ receptor in the dorsal raphe nucleus. Interestingly, these effects were associated with increases in anxiety-like behavior in juvenile primates [121,122]. In rodents, similar findings were obtained, since pregestational together with perinatal high-fat diet resulted in an increase in dorsal hippocampal BDNF and ventral hippocampal 5-HT_1A_ gene expression and in anxiety-like behavior in adult offspring. Although, the offspring conditioned fear response and exploratory behavior were not affected [123].

#### 3.2.3. Maternal Alcohol Consumption Reduces Fetal 5-HT Production 

Moderate to severe maternal alcohol consumption throughout pregnancy is associated with a disrupted brain development, attention deficits, hyperactivity, and impulsiveness in the offspring (see review [124]). Prenatal alcohol consumption is potentially linked to decreased 5-HT levels in maternal human serum and reduced 5-HTT binding in children’s medial frontal cortex [125,126]. Similarly, decreased rodent fetal whole brain 5-HT levels were reported, irrespective of the number of days exposed to alcohol [127,128,129]. The maternal alcohol-induced reduction in fetal 5-HT levels probably occurs due to reduced 5-HT-ergic neurons and TPH-positive cells within the raphe nuclei of the offspring [130,131,132,133,134]. These reductions have been linked to a lower expression of several pro-survival genes (i.e., the NF-κB-dependent anti-apoptotic genes: XIAP, cIAP1, cIAP2, Bcl-2, and Bcl-xl) and an increase in apoptotic cell death in the rat hindbrain [135,136]. Indeed, the migration and differentiation of 5-HT neurons towards their final position within the raphe nucleus was found to be diminished in the offspring by prenatal alcohol exposure [134]. Interestingly, such distortion in 5-HT-ergic neurons in the raphe nuclei could be prevented by maternal treatment with a 5-HT_2A/2C_ receptor agonist or a 5-HT_1A_ receptor agonist [127,133,136,137]. More specifically, the 5-HT_1A_ agonist, ipsapirone, which likely acts on somatodendritic 5-HT_1A_ receptors on both raphe neurons and astrocytes. Through agonistic activations of these cell types, ipsapirone prevented the reduction in expression of the pro-survival genes XIAP and Bcl-xl. Hence, these pro-survival genes might provide a mechanism into how ethanol-induced apoptosis can be protected via ipsapirone. [133,136]. Behaviorally, maternal alcohol-induced decrements in raphe 5-HT-ergic neurons resulted in a decreased susceptibility of and resistance to anxiety [131].

In rodents, maternal alcohol consumption not only results in alterations in 5-HT-ergic neurons, but also in 5-HT-ergic fibers from the raphe nuclei. More specifically, fewer 5-HT fibers were observed in E15- and E18-fetal’s brain in a variety of brain areas including the hypothalamus, hippocampus, and frontal and parietal cortices, and a reduction in the growth of the 5-HT-regulated somatosensory thalamocortical projections was observed [138]. Underdevelopment of the thalamocortical projections is thought to be the underlying cause of the volume reduction seen in the posterior medial barrel subfield (especially layer IV) and of the reduced numbers of neurons in this area. These alterations have been suggested to lead to compromised sensory modality [139]. Moreover, the 5-HTT gene expression and binding of the HPA-axis and prefrontal-limbic circuitry also seem to be sensitive to dysregulation evoked by prenatal alcohol exposure [140].

### 3.3. Maternal Stress Affects the 5-HT System, Neurodevelopment, and Behavior in the Offspring

Maternal stress (defined as physical and/or psychosocial stress during pregnancy) can negatively influence child health and a variety of pediatric aberrations have extensively been reported (for review [141]). The effects of maternal stress on the maternal HPA-axis resulting in excessive fetal cortisol levels has been most extensively described in literature. In recent reviews, the effects of maternal stress on the placental and fetal 5-HT systems is briefly discussed as potential underlying factor as well. These 5-HT alterations may affect fetal brain development and increase the risk of the onset of neuropsychiatric disorders, such as ASD, depression, and anxiety [141,142,143]. In the following sections, we will review human and animal studies describing the effects of maternal stress on 5-HT-related brain development and function in the offspring. In animals, stressors used include electric foot shocks, restraint under bright light, auditory stimuli, cold water immersion, forced swim, wet bedding, crowding, saline injection, and cage rotation. Some studies use one of these stressors to mimic maternal stress, while others make use of chronic unpredictable stress models whereby, in general, animals are exposed to up to two of these stressors daily. In short, maternal stress is associated with both increases and decreases in 5-HT levels in the offspring. Increases in 5-HT levels affect development of the thalamocortical circuitry, whereas decreases in 5-HT levels affect development of the prefrontal-limbic circuitry. The HPA-axis is affected in almost all maternal stress models, irrelevant of the offspring’s 5-HT level changes. 

#### 3.3.1. The Role of Prenatal Stress in Offspring Brain Development in Humans

Human studies clearly show that prenatal stress affects neurodevelopment including functional and structural brain connectivity alterations involving the amygdala and frontal cortex and changes in the HPA-axis functionality. In turn, these alterations induce long-lasting effects on children’s mental health (see review [144]). Studies concerning the role of maternal and fetal 5-HT levels in relation to stress, however, are scarce. Rotem-Kohavi and colleagues investigated neonatal brain function organization of children exposed to either maternal depression (indicative of maternal stress) only or selective serotonin re-uptake inhibitors (SSRIs) and maternal depression. Neonates from the depressed-only group showed higher hub values (hubs are highly connected brain regions) in the left anterior cingulate, insula, caudate, and amygdala. Hub values of these regions in neonates of SSRI-depressed mothers did not differ from neonates of healthy mothers. These findings may suggest that SSRIs normalize the depression-induced alterations during brain development [145]. Consequently, both maternal depression and prenatal SSRI-exposure might have overlapping underlying mechanisms, which potentially involves the fetal 5-HT system. Indeed, high maternal anxiety, depression, and anger scores during and after pregnancy decreased neonatal peripheral 5-HT levels together with greater relative right frontal EEG activation, altered sleep patterns, and lower motor organization. However, no difference in maternal peripheral 5-HT levels were found during and after pregnancy [146].

#### 3.3.2. Prenatal Stress Alters the 5-HT System in the Animal Offspring

During the embryonic period, maternal stress increases fetal brain 5-HT levels. An early study reported that chronic mild psychosocial prenatal stress increased maternal plasma tryptophan levels and increased levels of tryptophan, 5-HT, and 5-HIAA in fetal whole brain and neonatal cortex [147]. Maternal stress induced via either pregestational or prenatal chronic unpredictable stress led to increased forebrain, hippocampal, and hypothalamic 5-HT levels, decreased 5-HTT expression in both brain areas, and decreased hippocampal 5-HT_1A_ receptor activity in the fetus [148,149]. However, animal studies have reported conflicting data regarding maternal stress-induced lasting effects on offspring’s 5-HT levels. Researchers found decreases in brain 5-HT levels of adolescent and adult offspring when using different maternal stress models. Upon maternal stress, hypothalamic, hippocampal, and mPFC 5-HT levels were reduced and PFC 5-HT metabolism was increased [150,151,152,153,154,155]. TPH2 gene and protein levels were decreased in the dorsal raphe nucleus, and the hippocampus and 5-HTT protein levels were increased in hippocampal and PFC brain regions [154,156]. Another group of researchers showed data in favor of maternal stress-induced increases in adult offspring 5-HT levels in similar brain regions and using similar maternal stress models. Maternal stress resulted in a decreased hippocampal 5-HTT and alterations in the dorsal raphe, i.e., increases in 5-HT levels and TPH2 gene expression and enhanced TPH2 immunoreactivities [157,158]. Of note, another study showed prenatal restraint stress-induced decreases in 5-HT-positive cell density and overall 5-HT immunoreactivity in the dorsal raphe nucleus of adult animals but increases in overall immunoreactivity in the mPFC and increased hippocampal TPH2 immunoreactivity. No difference was found in TPH2 immunoreactivity in either the dorsal raphe nucleus or the mPFC [159]. 

#### 3.3.3. Prenatal Stress-Induced Alterations in Brain Circuits and Behavior in the Animal Offspring

The above-mentioned studies show not only alterations in offspring’s 5-HT levels in response to maternal stress but also imply neurodevelopmental changes. Prenatal chronic unpredictable stress not only increased embryonic forebrain 5-HT levels but also increased 5-HT+ fetal thalamic neurons and thalamocortical afferents [149]. Studies showing a decrease in an offspring’s 5-HT levels in response to maternal stress suggest alterations within the prefrontal limbic circuitry. Pregestational and prenatal stress reduced hippocampal neurogenesis and reduced synaptic plasticity (synaptophysin and PSD-95 densities) in the mPFC and hippocampus of adolescent rats [150,151,155,160]. Moreover, prenatal chronic unpredictable stress caused hippocampal structural modifications (decreases in ventral hippocampal sub-region volume together with a decrease in number of neurons; increases in dorsal hippocampal sub-region volume) [153]. Research supporting maternal stress-induced increases as well as decreases in an offspring’s brain 5-HT levels have suggested a maternal stress-induced over-activated HPA-axis in the offspring. Different maternal stress models increased fetal and adult offspring plasma and serum (baseline and stress-induced) corticosterone levels, corticotrophin releasing factor, and the hormone adrenocorticotropic and increased amygdala corticotrophin releasing factor gene expression [148,153,154,158,159]. Furthermore, diminished hippocampal glucocorticoid receptor levels and density were detected [158,160]. Of note, maternal stress-induced fetal changes in serum corticosterone and corticotrophin releasing factor have been suggested to cause the discussed 5-HT-ergic modifications. Whilst such changes in corticosterone in adulthood are proposedly caused by long-term changes in HPA function in response to the altered 5-HT-regulated cortical releasing factor neuronal activity [148].

Not only brain morphology but also behavior of the offspring is affected by maternal stress. Many studies show maternal stress-induced increases in offspring’s anxiety and depressive-like behavior, irrespective of the applied maternal stress model and 5-HT levels measured in the offspring [153,154,156,157,158,159,161,162,163]. Consistently, animals exposed to prenatal stress did not develop any form of stress adaptation [164]. However, in a few other studies prenatal stress did not affect or even decrease depressive- and anxiety-like behaviors [162,165]. Proposed mechanisms underlying these anxiety- and depressive-like behaviors are an increased HPA-axis reactivity, decreased transcriptional regulation factors CREB and BDNF in the prefrontal-limbic circuitry, and increases in placental inflammation and placental IGFBP-1 expression. Increases in this gene expression causes a decrease in available growth factors [153,154,158,161]. No clear association was found with hyperactivity or impulsivity, which are putative indicators for ADHD. Most studies did not find any effect, however one study reported hyperactive adult mice and two other studies reported hypo-locomotion [153,154,157,158,161,162,165]. Prenatal restraint stress and pregestational chronic unpredictable stress, respectively, decreasing and increasing offspring’s 5-HT metabolism, reduced social behavior in juvenile and adult offspring. Mechanisms underlying these behavior alterations probably include the above discussed reduced hippocampal plasticity and over-activated HPA-axis function [150,152,160]. Reduced social behavior has been suggested to be a potential indicative for the development of ASD-like behaviors [152]. 

### 3.4. Maternal Intake of 5-HT-ergic Medication Alters 5-HT Levels in Offspring and Affects Their Neurodevelopment and Behavior

Pregnant women suffering from depression, migraine, or schizophrenia are sometimes in need of medication during pregnancy. Medication is taken during 6% of all pregnancies [166]. Over the years, multiple reviews have been written trying to disentangle the effects attributable to prenatal SSRI exposure from the underlying maternal disorder and the maternal-child transmission of risk alleles to neuropsychiatric disorders. Due to inconsistent study results, review outcomes differ. There is a tendency, however, towards viewing SSRI exposure as a plasticity rather than a risk factor [167,168,169,170,171,172,173]. This means that SSRI exposure can positively or negatively influence offspring’s 5-HT system depending on other gestational factors. In the following sections, we describe the effects of maternal intake of 5-HT receptor (ant)agonists and SSRIs during pregnancy on offspring’s 5-HT-ergic brain development and neuropsychiatric disorders-related behaviors, referring to both human and animal studies. In short, there is quite some overlap between human and animal findings. Maternal SSRI intake alters fetal 5-HT levels; however, the direction remains unclear. Nevertheless, these fetal brain 5-HT alterations could explain the remarkable changes in fetal brain development related to the thalamocortical and prefrontal-limbic circuits, as well as the HPA-axis that we address in this review.

#### 3.4.1. Maternal Intake of 5-HT Receptor (Ant)Agonist Might Affect the Unborn Child

Triptans are 5-HT_1B/1D_ receptor agonists, which are often prescribed as medication against migraine and cluster headaches. Triptans cross the placenta [174] and, thus, may bind to these receptors in the fetal brain. Indeed, a clinical study showed an increased risk of attention problems in children prenatally exposed to triptans (particularly when exposed during the first trimester of pregnancy) [175]. Furthermore, antipsychotic medications, such as clozapine, have a high affinity as an antagonist/inverse agonist for (placental) 5-HT_1A_ and 5-HT_2A_ receptors [176,177]. Therefore, clozapine might be able to influence the placental or fetal 5-HT system and thus fetal neurodevelopment.

#### 3.4.2. Maternal SSRI Intake is Associated with Neural Changes and Behavior in Offspring in Humans

In many countries, provided guidelines prefer the use of psychotherapy and other non-pharmaceutical treatments as first-line treatments against mild to moderate depression during pregnancy. Antidepressants such as SSRIs are only preferred in severe cases of depression. However, current practice shows that most of the time antidepressants are still used as the primary treatment [178,179,180]. The effect of SSRIs during pregnancy on an offspring’s development has been examined frequently. Indications of a reduced brain maturation and neural development are the decrease in reelin and S100β found in infants of mothers exposed to SSRIs during pregnancy [181,182]. The programming of the HPA-axis seems to be altered as well, as reflected by increased serum corticosteroid binding globulin levels (even when controlling for maternal depression) and decreases in cortisol reactivity [183,184]. In support, Brennan et al. [184] suggested a moderate role for psychotropic medication (mainly SSRIs) in the relationship between maternal disorders during pregnancy and infant cortisol levels. Due to such changes, altered brain morphology is expected. Indeed, infants prenatally exposed to SSRIs have increased volumes of, and connectivity between, the amygdala and insular cortex [185]. Children prenatally exposed to SSRIs further displayed an increase in connectivity in the right medial frontal orbital gyrus [145]. Moreover, a recent study revealed that newborns prenatally exposed to SSRIs exhibited white matter changes in the basal ganglia and thalamus [186]. These white matter changes suggest that SSRI exposure affects axon myelination as well. As shown in Table 2, most human case-control and cohort studies have found an association between SSRI intake of the pregnant mother and risk of neuropsychiatric disorders (for more details see Appendix A). In general, studies consider the maternal neuropsychiatric condition as an important confounding factor. Interestingly, even though the association with offspring depression decreased, the association stayed significant when comparing SSRI-depressed mothers with depressed-only mothers instead with healthy controls [187,188]. As to whether prenatal SSRI exposure evokes a risk of ASD development is unclear. Positive associations between prenatal SSRI exposure and ASD are mainly found in meta-analyses focusing on case-control studies and on early gestational exposure (pre-gestation and first trimester). 

#### 3.4.3. Maternal SSRI Intake Affects Animal Offspring 5-HT Signaling and Behavior

Numerous developmental consequences of gestational SSRI exposure (i.e., prenatal, neonatal, or perinatal) have also been reported in animal studies. These investigations mainly used healthy animals. To translate animal findings to the clinic, it is imperative to study gestational SSRI exposure not only in healthy animals but also in combination with maternal stress during pregnancy. As discussed below, gestational SSRI exposure seems to be able to mitigate some of the developmental effects of prenatal stress. It is, however, also able to alter developmental mechanisms and behaviors without counteracting prenatal stress; similar to the human study of Rotem-Kohave et al. [145]. That both gestational SSRI exposure and prenatal stress can influence brain development independent of each other is also highlighted in another review focusing specifically on hippocampal plasticity [203].

Animal models reported behavioral changes that partly correspond to those found in human studies (Table 3 & Appendix A). Like in humans, rodents gestationally exposed to SSRIs do not seem to show increases in hyperactivity or impulsivity. As in human studies, several, but not all, studies do show an increase in anxiety- and depression-like behavior due to gestational SSRI exposure. There is no clear reason why studies using fluoxetine during a similar period and of similar concentrations evoke different depressive-like outcomes across studies. Different outcomes might be related to the specific SSRI, SSRI concentration, or behavioral test used. For instance, studies that tested citalopram or used the novelty suppressed feeding test (a test suggested to detect behaviors related to anxiety as well as depression [204]), mostly suggest an increased anxiety-like behavior. Studies using the highest fluoxetine concentrations resulted in decreased anxiety levels. Studies without significant differences mainly used the elevated plus maze as behavioral test (a test supposedly primarily detecting anxiety levels [204]) and low fluoxetine concentrations. Regarding ASD, rodent studies often investigate alterations in the sensory system, repetitive behavior, and social behavior (including social preference, social play, and social interaction). Gestational SSRI-exposed offspring mostly show an impaired social preference and social play behavior. Inconsistencies concerning social play and social interaction outcomes are probably due to environmental differences, i.e., semi-natural environment versus familiar versus unfamiliar playmates and juvenile versus adolescent versus adulthood. An earlier review of Gemmel et al. [151] suggested that the timing of gestational SSRI exposure might explain differences in social behavior. However, this is not visible when looking at the current available literature (Table 3). Nonetheless, overall, ASD-related behavior seems to be affected by SSRIs. 

A few studies investigated the effects of gestational SSRI exposure on the 5-HT system of the offspring. Offspring of pregnant mice that were perinatally treated with fluoxetine and sertraline or prenatally with escitalopram exhibited whole brain decreases in MAOA expression and whole brain increases in cortical expression of TPH2, 5-HTT, and 5-HT_1A/2A/C_ receptors [218,221,227]. These molecular changes likely lead to increased 5-HT signaling. However, other studies showed a reduction in brain 5-HT levels in the offspring. More specifically, perinatal fluoxetine resulted in decreased juvenile whole-brain 5-HT levels [165,220]. Neonatal escitalopram exposure led to decreased adult hippocampal 5-HT levels along with reduced 5-HTT binding in the median and dorsal raphe nucleus [212]. Notably, neonatal fluoxetine did not alter adult 5-HT levels in the mPFC, hippocampus, and somatosensory cortex [206,212]. Contradicting findings are also seen in studies investigating the effects of gestational SSRI exposure on prenatally stress exposed offspring. Prenatal citalopram exposure has been reported to normalize prenatal chronic unpredictable stress-induced increases in fetal forebrain 5-HT levels [149]. Neonatal fluoxetine exposure to prenatally restraint stressed offspring decreased hippocampal 5-HT levels [233]. Gestational SSRI intake also normalized maternal stress-induced decreases in an offspring’s 5-HT level. For instance, pre-adolescent offspring from dams pregestationally exposed to chronic unpredictable stress and perinatally treated with fluoxetine, reported that fluoxetine treatment normalized the diminished 5-HT levels and 5-HT metabolism in the PFC and the hippocampus [150,151]. 

#### 3.4.4. Maternal SSRI Intake Affects Animal Offspring’s Brain Circuitry Development

Changes in offspring’s 5-HT system may result in altered functioning of specific brain circuits. One of the circuits that is mainly affected in the offspring from neonatal SSRI exposed healthy dams is the thalamocortical circuitry. Alterations within this circuitry are associated with the above discussed sensory abnormalities and decreased social behavior [206,225,229]. Fluoxetine exposure led to a reduction in the complexity of thalamocortical afferents projecting to layer four of the barrel cortex (i.e., fewer/shorter branches) [225]. Further, paroxetine exposure was associated with altered refinement, but not the formation, of dense clusters of these afferents (i.e., reduced barrel size and enlarged septa) [234]. Neonatal citalopram exposure suppressed the amplitude and prolonged the delay of sensory-evoked potentials, reduced the power and frequency of the early gamma oscillations, and suppressed sensory evoked and spontaneous neuronal firing in the barrel cortex of neonatal rats. These effects were absent during adolescence, indicating that the changes were transient [235]. Finally, retrograde tracer studies in neonatal fluoxetine exposed rats revealed a change in the morphology of oligodendrocytes in the corpus callosum. This change was associated with altered axon myelination in the corpus callosum and reduced connectivity between the primary somatosensory cortices across the hemispheres [229]. Interestingly, prenatal citalopram intake was found to decrease p11 expression in fetal thalamic neurons. The authors suggest that this decrease in p11 expression alters 5-HT signaling through regulating the translocation and signaling of 5-HT_1B/D_ receptors, thereby potentially disturbing thalamocortical circuitry formation [236]. This might explain how prenatal citalopram exposure decreases and, therefore, normalizes the increased 5-HT-containing thalamocortical afferents evident in prenatal stressed fetus [149]. Thus, while neonatal SSRI exposure in healthy animals negatively influence thalamocortical circuitry development, these alterations seem to normalize the gestational stress-induced effects.

The HPA-axis is a brain circuitry, which is mainly affected by gestational SSRI intake studied in the offspring from stressed dams. Alterations have been related to the earlier discussed increases in anxiety-like behavior (probably via increased 5-HT_2C_ receptor gene expression in the PFC and amygdala) and enhanced social preference [213,232]. While prenatal fluoxetine in healthy dams reduced stress-induced plasma corticosterone levels in adult male offspring, neonatal fluoxetine in healthy dams increased these levels in the serum of adult male offspring. In both studies, basal corticosterone levels were unaffected [213,222]. Consistently, when studying perinatal fluoxetine in prenatal restraint stressed offspring, fluoxetine normalized prenatal stress-induced increases in HPA-axis reactivity in adulthood. This time, however, perinatal fluoxetine also decreased baseline serum corticosterone levels [161]. Such a decrease in plasma basal corticosterone levels was also found in adolescent male rats after neonatal fluoxetine exposure. This time, however, prenatal restraint stress did not affect these levels [237]. That perinatal SSRI exposure, irrespective of prenatal stress, can impact the HPA-axis has been shown in other studies as well. Perinatal fluoxetine resulted in increased serum corticosteroid binding globulin, decreased hippocampal synaptic protein (PSD-95), reduced glucocorticoid receptor density in the hippocampus, and reduced glucocorticoid receptor density in the mPFC [150,160,203,237].

Another brain circuitry affected by gestational SSRI exposure in both healthy and prenatally stressed animal models is the prefrontal-limbic circuitry. A few studies suggest a link between presynaptic 5-HT_1A_ activity and social interaction or depressive-like behavior and between hippocampal neurogenesis and anxiety-like behavior [210,213,218]. Other studies propose the decrease in mPFC dendritic complexity as one of the underlying mechanisms in decreased social recognition and increases in anxiety-like and depressive-like behaviors [7,206,216,223]. Neonatal fluoxetine did not affect adult hippocampal neurogenesis but normalized decreased hippocampal neurogenesis evident in prenatally restraint-stress adolescent offspring [162,213]. Reduced adult hippocampal neurogenesis in males from pregestationally stressed dams was not normalized by perinatal fluoxetine. Moreover, females from this study showed no changes due to maternal stress but did show increased adult hippocampal neurogenesis as a consequence of the perinatal fluoxetine exposure [160]. Another study showed increases in prefrontal-striatal connectivity/synchronization after prenatal exposure to citalopram. Underlying mechanisms may include a decrease in the complexity of prefrontal neurons (i.e., fewer/shorter branches and reduce PSD-95), supposedly via 5-HT_3_-regulated increases in reelin [7,160,216,238]. Of note, other studies have reported an increased synaptic plasticity [150,206]. Another mechanism suggested is the disturbed mPFC excitation/inhibition as determined by a downregulation in NMDAR1 and CaMKIIα expression and an increase in GABAergic interneurons in the mPFC. The changes related to the mPFC excitation/inhibition might again be a compensation for the overall increased prefrontal excitation [223,238]. In support, prenatal fluoxetine exposure caused augmented spontaneous inhibitory synaptic transmission onto the layer 5 pyramidal neurons within the mPFC and led to an increase in the migratory speed of inhibitory cortical interneurons [239]. This is probably due to an upregulation in the 5-HT_2A_ receptor signaling or may be related to a reduced expression of the transcription factor Npas4 [223,240]. Interestingly, alterations in the pyramidal neuron excitability seem to depend on the location. That is, an increased excitability was found in the prelimbic cortex, while in the infralimbic cortex a decreased excitability was found [7]. Adult male offspring neonatally exposed to fluoxetine showed an increase in the density of hippocampal immature neurons. In adult female offspring a decreased density was found [213]. Moreover, perinatal fluoxetine exposure resulted in decreased perineuronal net formation at postnatal day 17 (ongoing critical period [7]) and 24 (critical period closed [7]) in offspring’s hippocampus and basolateral amygdala [241]. Given that perineuronal nets increase with the maturation of neurons, the latter finding suggests that prenatal SSRI exposure delays the onset and the closure of a critical period in the development of the hippocampus and amygdala. Lastly, pregestational fluoxetine exposure affected 5-HT_2C_ receptor gene expression and editing in neonatal amygdala and PFC. Maternal restraint stress prior to pregestational fluoxetine administration revealed that fluoxetine exposure can either reverse or enhance 5-HT_2C_ receptor editing in the neonatal PFC [232]. 

### 3.5. Maternal Immune Activation Affects the Tryptophan Pathway and Neurodevelopment of Offspring

Recent reviews have proposed a role for maternal immune activation in the aetiology of ADHD, ASD, and depression and suggest a potential mediating role for the 5-HT system [242,243]. Immune activation during pregnancy occurs either upon a viral or a bacterial infection, or when there is a condition such as pre-eclampsia [244]. Pre-eclampsia is a complex multisystem disorder unique to the second half of pregnancy and marked by low platelet 5-HT levels [245]. Pre-eclampsia has been shown to play an important role in the development of neuropsychiatric disorders such as ASD [246,247,248]. In the following sections, we describe the effects of induced maternal immune activation on 5-HT-ergic brain development of the offspring of rodents. In short, maternal immune activation causes a decrease in an offspring’s brain 5-HT levels throughout life, independent of the method used and gestational period studied. During the embryonic period an acute increase in brain 5-HT levels may occur. The tryptophan and 5-HT-related brain areas that seem to be affected by maternal immune activation are located in the thalamocortical and the prefrontal-limbic circuits. 

#### 3.5.1. Activation of the Fetal Immune System Influences Animal Offspring

Experimentally, the consequences of maternal immune activation in fetal brain development are often studied in rodents using the immune stimulant polyriboinosinic–polyribocytidylic acid (Poly(I:C)), an endotoxin injection (*Escherichia coli* injection) or a flu exposure (influenza administration). Poly(I:C) affects maternal cytokine signaling, including interleukins (IL) such as IL-6. Since IL-6 seems to be able to pass the placenta, it might be able to affect fetal brain development [249,250]. It is noteworthy that not all studies found evidence of IL-6 placental–fetal transport [251]. IL-6 is able to trigger fetal inflammatory processes both directly via placental transfer and indirectly via placental inflammation [252]. This, in turn, can have widespread effects on brain development, including a decrease in the survival rate of fetal rostral raphe 5-HT neurons [253]. 

#### 3.5.2. Changes in Placenta-Derived 5-HT Levels Influence the Animal Offspring

Besides cytokine-induced fetal 5-HT system alterations, maternal immune activation may also result in other 5-HT system alterations in either the mother or the placenta thereby affecting the offspring’s (neuro)development. A recent mouse study showed that even though Poly(I:C)-induced maternal immune activation was associated with a transient IL-6 increase in the maternal serum, there was no evidence of cytokine accumulation in the fetal brain [254]. Early-gestation poly(I:C) exposure evoked a transient increase in placental tryptophan levels and TPH1 gene expression and an increase in enzymatic activity. Placental MAOA gene expression was not affected [254]. Interestingly, when inducing maternal immune activation at mid- or late-gestation, the development of 5-HT-ergic neurons in the fetal hindbrain was not influenced [254,255]. This lack of effect is most likely because the hindbrain is not dependent anymore on placenta-derived 5-HT after E10.5. In contrast, E15–E17 endotoxin exposure did decrease dorsal raphe TPH2 neurons numbers and size when the offspring was investigated in adulthood [256]. 

A few studies investigated the effect of maternal immune activation on the fetal brain 5-HT system but obtained ambiguous results. Early-gestation-induced maternal immune activation increased 5-HT fetal forebrain levels, increased 5-HT-ergic neurons in the hindbrain, and changed forebrain circuitry formation (i.e., a reduction in 5-HT axon outgrowth into the forebrain) [254,257]. Late-gestation endotoxin injection in rats did not affect fetal cortical 5-HT levels but decreased fetal brain TPH1 gene expression [256]. Importantly, human intrauterine bacterial infection, as well as rodent-induced maternal immune activation, increased the placental kynurenine/tryptophan ratio [254,258,259]. In addition, the fetal rodent brain levels of kynurenine and its metabolites, quinolinic acid and kynurenine acid, were increased [254,258]. Thus, maternal inflammation may shunt placental tryptophan metabolism away from 5-HT to the kynurenine pathway. 

#### 3.5.3. Maternal Immune System Activation Influences Brain Circuits and Behavior in Animal Offspring

Multiple studies investigated the lasting effects of maternal immune activation on an offspring’s brain 5-HT system. Together, these studies show that, regardless of the gestational period, induction method, and animal’s age and species, maternal immune activation decreases 5-HT levels in the offspring (blood serum [260]; cerebellum [261,262]; frontal and parietal cortices, and the hippocampus [255,256,257,263]). This change was accompanied by a decrease in whole brain TPH2 and 5-HTT gene expression and an increase in the gene expression of TPH1 [256]. Early-gestation poly(I:C) did not affect 5-HT levels in the PFC, amygdala, ventral tegmental area, and the substantia nigra pars compacta [257,263]. Interestingly, while total striatal 5-HT levels were unaffected [257], subdividing the area showed a reduction in 5-HT and 5-HIAA levels in the nucleus accumbens but not in the caudate putamen [263]. On the contrary, poly(I:C) exposure at E15 decreased 5-HT and 5-HIAA levels in the caudate putamen but not in the nucleus accumbens of adolescent and adult offspring [264]. 

Beside the discussed decreases in an offspring’s brain 5-HT levels, these studies also reported changes in brain and behavior. Late-gestation maternal immune activation can lead to excitotoxic injury, such as increased apoptosis in the ventrobasal thalamus and a disrupted thalamocortical development in newborn pups (i.e., a decrease in 5-HT-mediated thalamocortical fibers and a decrease in 5-HTT expression in the somatosensory cortex) [255]. Early- to mid-gestation maternal immune activation-induced decreases in an offspring’s brain 5-HT levels were paralleled by sensory abnormalities and a reduced social contact [260,264]. Additionally, male offspring showed increased locomotor and stereotypic behaviors. As suggested by the author, these findings might be indicative of the development of ASD-like behaviors [260]. Both early- and late-gestational activation of the maternal immune system causes decreases in an offspring’s brain 5-HT levels and results in anxiety-like behavior [256,260]. Offspring gestationally exposed to poly(I:C) stimulation, specifically during mid-gestation, showed an increase in depressive-like behaviors together with increased hippocampal 5-HTT levels [265].

## 4. Discussion

### 4.1. Gestational Factors Influence Placenta-Derived and Raphe-Nuclei-Produced 5-HT Content

In the main body of this review, in Section 3, we described how five gestational factors affect the 5-HT system, the brain development, and the behavior in the offspring. In short, all reviewed gestational factors influence the 5-HT systems, by either increasing or decreasing 5-HT levels in the mother, the placenta and in the fetus, resulting in a risk of neuropsychiatric disorders in the offspring. In this Section, 4.1, we summarize the main findings related to the gestational factor influences on an offspring’s 5-HT system, and we discuss potential underlying mechanisms (Table 4). Subsequently, in Section 4.2, we discuss how the interaction between gestational factors might amplify or counteract each other’s impact on the fetal serotonergic system. Next, we summarize and discuss in Section 4.3 the main findings related to the gestational-factor-induced 5-HT-ergic alterations of the structure and function of the thalamocortical areas, the prefrontal-limbic areas, and the HPA axis. Lastly, in Section 4.4, we report the main findings related to the gestational-factor-induced risk of neuropsychiatric disorders, and we discuss the potency of 5-HT-ergic dependent alterations as underlying mechanisms within the mentioned circuits.

The influence of maternal 5-HT-ergic genotype depends on the gene investigated and whether the gene variance involves a loss- or gain-of-function mutation. Maternal genotype might influence the fetal 5-HT system through a variety of mechanisms altering the placenta-derived 5-HT content. The genotype of the mother may influence maternal gastrointestinal tract function [266] or its responsivity to gut bacteria (i.e., microbiome [267]). Importantly, the gastrointestinal tract and microbiome, in turn, influence maternal blood tryptophan and platelet 5-HT levels [267,268]. Consequently, changes in maternal tryptophan or 5-HT levels potentially alter 5-HT uptake into, and transfer through, the placenta. Maternal genotype may affect placental 5-HT synthesis (in cases of TPH1 and/or TPH2 polymorphisms) and placental immune- and metabolic-related activations (in case of maternal 5-HTT availability [6]). In turn, affecting placental 5-HT homeostasis. As maternal risk alleles can be transferred to the offspring, maternal genotype is also able to influence offspring’s 5-HT system directly (i.e., 5-HT production, availability, and receptor activity).

Regarding 5-HT-related maternal diet, tryptophan-deprived diets, high-fat diets, and alcohol consumption seem to mainly reduce an offspring’s 5-HT levels. Instead, high-tryptophan intake and protein-restricted carbohydrate-enhanced diets seem to be associated with increased 5-HT levels in offspring. Importantly, for some of the diets, the direction of change seems to differ depending on the time measured (embryonic vs. throughout life). Longitudinal studies are highly advised to investigate this research question. Potential mechanisms underlying maternal diets that are specific for tryptophan alterations and protein-restricted carbohydrate enhancements are changes in maternal tryptophan and 5-HT levels and their placental uptake and, consequently, the transfer to the fetus. The maternal alcohol consumption-induced reduction in fetal brain 5-HT levels might be a consequence of alterations within the intestinal microbiota composition and their function [269,270]. In turn, as shown for the maternal genotype, these alterations can influence maternal blood tryptophan and platelet 5-HT levels [267,268]. A similar mechanism can be suggested for the maternal high-fat diet-induced changes in offspring brain 5-HT levels [271,272]. 

Maternal stress increased fetal brain 5-HT levels in the few studies investigating the 5-HT levels during the embryonic period. After birth, both increased and decreased 5-HT levels have been reported after maternal stress. Although there are many differences between maternal stress models, even studies with similar models, time-windows gender, or ages find opposing effects of maternal stress on the offspring’s 5-HT system (e.g., regarding 5-HT levels and the expression of TPH2 and 5-HTT; Appendix A). This suggest that these contradicting results in 5-HT levels after birth are due to complex underlying mechanisms. We would like to highlight the maternal stress-induced decrease in the expression of Lmx1b, a transcription factor essential for central 5-HT neurons differentiation and maintenance in the fetal hindbrain and adult raphe nuclei [164]. The decrease in this transcription factor could potentially lead to decrements in the raphe 5-HT system development and subsequently diminished brain 5-HT production. Increases in offspring’s brain 5-HT levels are hypothesized to be related to alterations in placenta-derived 5-HT. Increased maternal tryptophan might either be transported directly to the fetus or may increase maternal or placental 5-HT levels [142]. Alterations in the developing raphe nuclei system and the transient 5-HTT expression might be a compensatory mechanism of increased placenta-derived 5-HT content.

It is generally believed that maternal SSRI intake increases fetal 5-HT levels (see review [273]). Many of the behavioral and mechanistic neurodevelopmental effects of gestational SSRI exposure resemble those observed in rodents lacking the 5-HTT [29,239,240,274,275,276]. Since 5-HTT knockout in rodents is associated with increased extracellular levels of 5-HT [277], it can be speculated that gestational SSRI exposure also increases fetal 5-HT levels. However, the few studies that investigated the actual 5-HT levels of the offspring are inconsistent and cannot be explained by the use of different types of SSRIs nor by differences in exposure periods. Interestingly, studies investigating both gestational stress and SSRI exposure mainly show that SSRI exposure normalizes the prenatal stress-induced effects on offspring’s 5-HT levels, irrespective of whether prenatal stress increases or decreases 5-HT levels in the offspring (Appendix A). Maternal 5-HT-ergic medication intake can influence the fetal 5-HT system through multiple mechanisms. First, medication decreases maternal platelet 5-HT levels [278,279], which potentially decrease 5-HT placental transfer. In addition, since maternal blood platelets and the placenta both express 5-HTT [280,281], medication such as SSRIs may affect placental function as well as 5-HT transfer. Moreover, most 5-HT-ergic medications are able to pass through the placenta, thereby creating the possibility to bind to the fetal 5-HTT and influence the offspring’s brain 5-HT receptor expression [282,283]. 

Maternal immune activation decreases the offspring 5-HT levels after birth, independent of the method used and gestational period induced. However, measurements during fetal development are ambiguous and may suggest a temporal increase in 5-HT levels. Placenta-derived cytokines are able to activate fetal inflammation, which might affect 5-HT-positive cell survival in the raphe nuclei. High maternal IL-6 has been suggested to increase placental A/L system amino acid transport [284] and, thus, potentially increases placental tryptophan transfer and thereby also fetal brain 5-HT levels. On the contrary, a decrease in placenta-derived 5-HT levels may be because pro-inflammatory cytokines in the maternal blood activate IDO and TDO, which in turn, trigger the production of kynurenine. In this way maternal inflammation may shunt placental tryptophan metabolism away from 5-HT to the kynurenine pathway. In support, maternal immune activation increased IDO expression in the placenta and in the fetal periventricular region together with an intense microglial activation within the fetal periventricular region [254,258].

### 4.2. Synergistic Impact on 5-HT-ergic Systems and Fetal Neurodevelopment

As shown in Figure 2, the discussed gestational factors do not only independently affect the 5-HT system, and thereby the brain development of the offspring, but also in interaction. As discussed in Section 3.2.2, high-fat diets leading to obesity during pregnancy increase the risk of maternal and placental immune activation through increased release of pro-inflammatory cytokines [117,120] but also the risk of pre-eclampsia via reduced placental vascularity and blood flow leading to a restriction in placental nutrient delivery [284,285]. Similarly, maternal stress provokes an increase in maternal immune activation, as shown by human plasma and mice placental IL-6 increments [286,287]. Thus, although some interactions between maternal factors might result in logical consequences on the developing 5-HT system, other effects of maternal factor interactions are not that natural and need more investigation in future studies. 

As described in Section 3.3, Section 3.4 and Section 3.5, the three gestational factors of maternal depression, maternal SSRI intake, and pre-eclampsia can alter placental and fetal 5-HT levels and increase risk of ASD. These factors may interact, given that in humans prenatal antidepressant intake by depressed pregnant women increases the risk of pre-eclampsia [288,289,290]. Interestingly, this risk increases when SSRI treatment continue in the second trimester [288,291]. Of note, another study was unable to replicate these findings [292]. Importantly, a combined exposure of these three gestational factors may amplify their actions on the maternal and fetal 5-HT system and might increase the risk of the development of ASD in the offspring. The synergistic effect of prenatal SSRI intake and pre-eclampsia can be explained by SSRI-induced increases in placental 5-HT_2A_ receptor activity [26]. As the 5-HT_2A_ receptor is a key regulator of placentation (influencing placental growth and function), it is suggested to play a role in the pathophysiology of pre-eclampsia [293,294], thereby influencing the maternal immune system activity. However, the association between pre-eclampsia and 5-HT_2A_ receptor function was not supported in another study [295]. Moreover, it is hypothesized that factors including maternal diet and genetics can influence the 5-HT_2A_ receptor promoter [296]. Hence, maternal diet and genetics might influence the synergistic effect of prenatal antidepressant intake and pre-eclampsia. 

Previous studies have demonstrated that SSRIs have reduced efficacy in humans with low 5-HTT expression due to an inherited 5-HTTLPR S-allele [297]. As such, carrying the 5-HTTLPR S-allelic variant could be a factor providing some protection against the unwanted effects of SSRIs on an offspring’s neurodevelopment. If so, the mother may simultaneously have less benefit from the treatment, increasing the impact of maternal stress and depression on the fetal development. The 5-HTTLPR has also been suggested to affect stress tolerance. Specifically, 5-HTTLPR-SS mothers of ASD children reported more stressors and an increased reactivity to these stressors in comparison to 5-HTTLPR-LL mothers. This suggests that this specific maternal genotype together with prenatal stress increases offspring’s risk of ASD [298]. In support, mouse studies showed that a combination of maternal stress and maternal 5-HTT genotype affects epigenetic mechanisms in the embryo resulting in social deficits but not anxiety-like behavior [299,300]. Moreover, maternal TPH2 functional haplotypes influence maternal depression during and after pregnancy [301].

Maternal 5-HT-ergic genotype can affect the amount of tryptophan and 5-HT that is released into the maternal and placental circulation (i.e., altered gastrointestinal tract function or responsivity to the gut microbiome). In this way, maternal genotype may influence the impact of the other gestational factors. The described examples of interactions between maternal factors might be altered by the potential of the maternal diet composition to impact a potential negative effect of other maternal factors. A protein-rich carbohydrate-low diet potentially decreases the transport of tryptophan, while a protein-low carbohydrate-high diet seems to increase this transport. As a result, a protein-restricted carbohydrate-enriched diet or tryptophan-high diet may increase the reduced brain 5-HT levels evident in offspring exposed to the maternal alcohol consumption (3.2.3) and maternal immune activation (3.5). Further exploration of this potential approach is needed. It is imperative to keep in mind that both too low and too high 5-HT levels negatively affect fetal neurodevelopment.

### 4.3. Gestational Factors Influence Offspring’s Brain Circuits

There seems to be a relationship between maternal, placental, and fetal 5-HT system changes during neurodevelopmental events and the formation and function of the three considered brain circuits we have discussed. The thalamocortical circuitry is commonly targeted by (1) the maternal 5-HT-ergic genotype, (2) alcohol consumption, (3) stress resulting in increased offspring’s brain 5-HT levels, (4) SSRI intake, and 5) late-pregnancy immune activation. While a reduction in fetal 5-HT levels (maternal 5-HTT genotype, alcohol consumption, prenatal SSRI exposure, and immune activation) resulted in a loss or a broadening of thalamocortical afferents, an increase in 5-HT fetal levels (maternal stress & potentially neonatal SSRI intake in healthy dams) resulted in an increase in 5-HT-containing thalamocortical afferents or fewer branches. Interestingly, these findings may indicate that embryonic 5-HT alterations (prenatal SSRI exposure) result in opposing effects on thalamocortical development in comparison to the direct effects on an offspring’s 5-HT system seen after neonatal SSRI exposure.

The prefrontal-limbic circuitry contains brain functions, which are also often influenced by gestational factors. This circuitry is targeted by (1) stress models resulting in reduced offspring’s brain 5-HT levels, (2) SSRI intake, (3) immune activation, and (4) potentially by all discussed maternal diets. Maternal diets are associated with the offspring 5-HT system alterations in prefrontal-limbic subareas primarily. Focusing on hippocampal neurogenesis, it has been shown that maternal stress reduces this process while maternal SSRI exposure is often reported to increase hippocampal neurogenesis. Maternal SSRI exposure can, therefore, potentially adjust the maternal stress-evoked effects. However, SSRI-induced changes in hippocampal neurogenesis seem to differ depending on gender and age. Consequently, in some cases perinatal SSRI exposure alters hippocampal neurogenesis negatively. Furthermore, structural plasticity was reduced in offspring when exposed to either tryptophan-deprived diets, maternal stress, or SSRI intake by healthy dams. Interestingly, both tryptophan-deprived and maternal stress exposures were associated with decreased 5-HT levels in the offspring’s prefrontal-limbic areas. 

The HPA-axis is affected by fetal 5-HT changes via influences from (1) maternal stress resulting in increased and decreased offspring’s brain 5-HT levels, (2) SSRI intake, and (3) potentially by maternal 5-HT-ergic genotype, protein-restricted carbohydrate-enhanced diets, and alcohol consumption. Maternal genotype and diets may affect the HPA-axis via influencing 5-HT-ergic receptor expressions. Maternal stress and SSRI intake have a more widespread effect by altering not only receptor expression but also by altering the negative feedback loop through, amongst others, corticosterone/cortisol levels and glucocorticoid receptors in the hippocampus. Maternal stress, regardless of the direction of altered brain 5-HT levels, seems to cause a hyper-reactive HPA-axis. On the other hand, most studies concerning perinatal SSRI exposure (human, healthy, and perinatally stressed animal studies) indicate a decrease in HPA-axis reactivity, thereby normalizing the maternal stress evoked effects. Importantly, perinatal SSRI exposure also impacts the HPA-axis without normalizing maternal stress (e.g., reducing glucocorticoid receptor density in different brain areas). 

### 4.4. Gestational Factors Can Influence the Risk to Develop Neuropsychiatric Disorders

Gestational-factor-evoked alterations in the offspring’s brain 5-HT system seem to correlate with an increased risk in the onset of certain neuropsychiatric disorders (Figure 3). For instance, maternal TPH1 loss-of-function mutations have been associated with an increased risk of developing ADHD. As maternal TPH1 mutations alter embryonic cell proliferation rate and cause abnormalities in the development of embryonic cortical regions, these alterations may underlie the increased risk of ADHD. It can be speculated that these effects are caused by diminished fetal brain 5-HT levels, as TPH1 mutations probably result in lower maternal platelet and placenta 5-HT levels and, thus, a diminished exogenous 5-HT content. Furthermore, attentional problems were seen when exposed to Triptans, a 5-HT_1B/1D_ receptor agonist. Indeed, genetic mutations within these receptors have been associated with the risk of ADHD [302,303].

Regarding ASD, an increased risk was reported in children of 5-HTTLPR-LL-carrying mothers and in animals exposed to maternal stress or maternal immune activation. Interestingly, enhanced 5-HTT expression seems to result in a decrease in fetal forebrain 5-HT levels and alterations within the thalamocortical circuitry (5-HTT gain-of-function mutation and prenatal SSRI exposure). These findings may suggest a potential mechanism through which maternal 5-HTT genotype influences the onset of ASD. A hyper-reactive HPA-axis and a reduced structural plasticity within the prefrontal-limbic circuit are suggested as underlying mechanisms after exposure to maternal stress and SSRI intake. Importantly, there are a few indicators suggesting that maternal SSRI intake after maternal stress restores the stress-induced changes in the HPA-axis. More elaborated studies are highly advised to investigate this potential further. Still, both human and animal work concerning SSRI exposure is very inconsistent but overall suggests a risk of ASD, irrespective of the effects of prenatal stress. This association, however, might change depending on the environment (e.g., maternal 5-HT-ergic genetic variance affects fetal 5-HT levels, maternal predisposition to neuropsychiatric disorders, and SSRI efficacy). It would be of importance to take such environments into consideration as potential interacting factors. 

Increased anxiety-like behavior in the offspring is reported after gestational exposure to maternal genotype, high-fat diets, alcohol consumption, stress, SSRI intake, and immune activation. Research outcomes may imply that the SSRI-exposure-induced behavior is more related to depressive instead of anxiety-like behavior. Interestingly, high-fat diets, alcohol consumption as well as maternal immune activation are proposed to result in decreased 5-HT levels in offspring, which may imply a relationship between high anxiety levels and gestational-factor-induced decreases in an offspring’s 5-HT levels. Interestingly, these factors (including stress models resulting in reduced offspring’s brain 5-HT levels) are associated with a disrupted structure and function of the prefrontal-limbic circuitry. A hyper-reactive HPA-axis has been proposed as an underlying mechanism as well.

Maternal immune activation, stress as well as SSRI intake during pregnancy are associated with an increase in depressive-like behavior in the offspring. Interestingly, thus far only one animal study investigated prenatal SSRI exposure, rather than perinatal or neonatal exposure, and did not find any changes in depressive-like behavior [304]. This is consistent with findings from a human study showing that only late pregnancy SSRI exposure was associated with increased risk of depression [187]. These findings suggest that SSRI-induced effects on an offspring’s 5-HT-ergic system are more important than placental-derived 5-HT content.

## 5. Conclusions

Addressing both human and animal studies, we show that all gestational factors considered in the present review (maternal 5-HT-ergic genotype, 5-HT-related diet composition, stress, 5-HT-related medication, and immune activation) affect both placenta-derived 5-HT and an offspring’s 5-HT production and availability. Due to these 5-HT changes, all discussed gestational factors are able to alter thalamocortical and/or prefrontal-limbic circuits and/or the HPA-axis and alter neuropsychiatry-related behaviors. Therefore, 5-HT changes induced by gestational factors may provide a risk of the onset of disorders like ASD, ADHD, anxiety, and depression. To date, research does not provide an answer to which extent placenta-derived 5-HT changes and direct changes to an offspring’s 5-HT system are causing the offspring’s brain and behavioral changes. Studies comparing first (placenta-derived 5-HT content) versus third trimester of pregnancy (offspring’s brain 5-HT production) can be helpful. SSRI-induced normalizations of abnormalities within the prefrontal-limbic and thalamocortical circuits evident in prenatally stressed offspring might be related to contrasting effects of SSRIs and prenatal stress on brain 5-HT levels in the offspring. Thereby, illustrating the potential of SSRIs to be a plasticity factor (suggested in the review [305]). Inconsistencies to whether gestational SSRI exposure influences ASD, anxiety, and depression still exist, which may imply the influence of other gestational factors. We argue that beyond the understanding of how a single gestational factor affects brain development and behavior in the offspring, it is also critical to elucidate the consequences of interacting factors. Hence, we want to advocate to take gestational factors into consideration when performing human cohort studies to reveal risk factors for neuropsychiatric disorders. Concerning animal studies, we want to emphasize the importance to study the potential effect of combining different gestational factors. Indeed, there seems to be a shift towards research investigating maternal SSRI intake in combination with prenatal stress. Studying the combined effects of multiple factors creates the opportunity to examine, for example, whether protein-reduced carbohydrate-enriched maternal diets can reverse a reduced fetal brain 5-HT content due to a genetic mutation. Furthermore, developing and optimizing current methodologies, such as whole-genome mapping, may initiate opportunities to examine the contributions of maternal genotype effects beyond the transmission of risk alleles [306,307]. Such efforts can create the possibility to disentangle the underlying mechanisms of an increased risk of neuropsychiatric disorders in children who, for instance, had been exposed to SSRIs during their development. In conclusion, an enhanced understanding of the interacting gestational factors, and on which brain circuits they act, will increase our abilities to prevent pregnancies with disadvantageous consequences for the offspring.

## Figures and Tables

**Figure 1 ijms-21-05850-f001:**
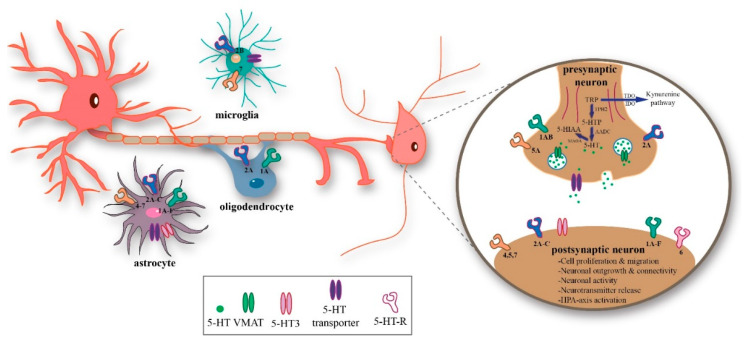
An integrated view of 5-HT signaling within the brain. 5-HT in the brain is synthesized from tryptophan via the enzymes Tph2 and AADC. 5-HT is transported via VMAT into synaptic vesicles. Exocytosis of these vesicles releases 5-HT into the synaptic cleft. Here, it can bind to 5-HT receptors localized on pre- and postsynaptic neurons and/or on glial cells (microglia, astrocytes, and oligodendrocytes). 5-HT signaling can be terminated by the reuptake of 5-HT via the 5-HT transporter. 5-HT is again transported into synaptic vesicles or converted to 5-HIAA via the enzyme MAOA. Within neurons, microglia, and astrocytes, tryptophan can be converted into kynurenine via the enzymes IDO and TDO. 5-HIAA: 5-hydroxyindole amino acid; 5-HT: serotonin; AADC: aromatic L-amino acid decarboxylase; IDO: indoleamine 2,3-dioxygenase; MAOA: monoamine oxidase A; TDO: tryptophan 2,3-dioxygenase; TPH2: tryptophan hydroxylase 2; VMAT: vesicular monoamine transporter.

**Figure 2 ijms-21-05850-f002:**
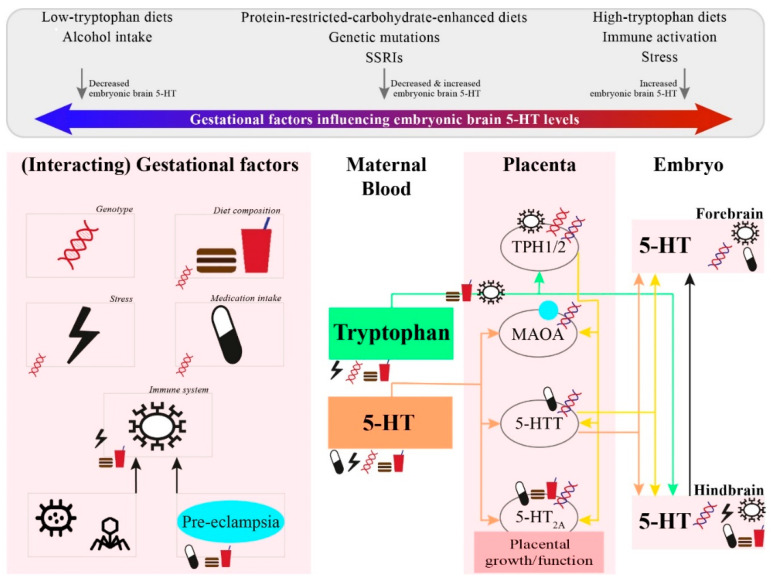
Schematic overview of the effects of (interacting) gestational factors on 5-HT levels during the embryonic period. Gestational factors (i.e., maternal 5-HT-ergic genotype, 5-HT-related diets, stress exposure, 5-HT-ergic medication intake, and immune activation) decrease (top arrow blue, left) or increase (top arrow red, right) embryonic brain 5-HT levels. The gestational factors in the middle (purple) have been found to increase as well as decrease brain 5-HT levels in the embryo. Gestational factors do not only independently affect the embryonic brain 5-HT system but are also able to interact and thereby amplify or counteract each other’s impact on the embryonic brain 5-HT system. Maternal genotype is able to influence the effects of maternal diet composition, stress, and medication intake. The maternal immune system is activated by bacteria, viruses, and pre-eclampsia (blue dot) and is influenced by maternal stress and diet composition. Furthermore, the risk of pre-eclampsia is influenced by maternal medication intake and diet composition. Gestational factor influences on the embryonic brain 5-HT system are a consequence of changes in placenta-derived (maternal blood or placenta) and/or offspring’s raphe-produced (hindbrain) 5-HT content. Factors can influence maternal tryptophan levels, which influence tryptophan transfer to the embryo and placental 5-HT synthesis (green arrows). In turn, placental 5-HT synthesis alterations influence the transfer of 5-HT to the embryo (yellow arrows). Maternal 5-HT levels can influence placental 5-HT levels and transfer to the embryo as well (orange arrows). It should be mentioned that the placenta also express MAOA and the 5-HT_2A_ receptor, which can breakdown the supplied/synthesized 5-HT and affect the placental structure and function, respectively. Gestational factors can influence the placental 5-HT system by targeting TPH1/2, MAOA, 5-HTT, and/or the 5-HT_2A_ receptor. Some of these factors also influence the embryonic brain 5-HT system. These direct effects are evident in the fore- and hindbrain and include fetal 5-HT-ergic genotype, the placental transfer from 5-HT-ergic medications, activation of the embryonic immune system, and alterations in the raphe-produced 5-HT content due to maternal stress (placental CORT transfer) and maternal diet. These factors can influence the maternal and placental pathway at multiple levels: altering maternal tryptophan and 5-HT levels, placental 5-HT synthesis, or the placental transfer from tryptophan and 5-HT to the fetus.

**Figure 3 ijms-21-05850-f003:**
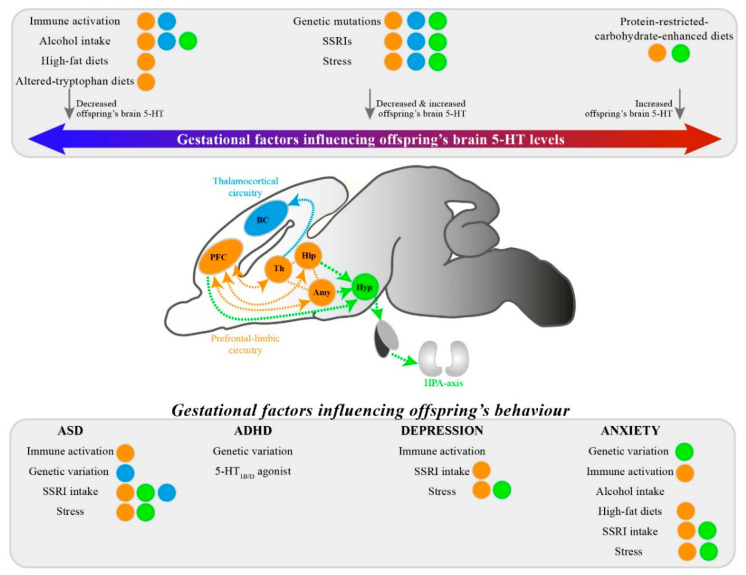
Schematic overview of the effects of gestational factors on the offspring’s 5-HT-ergic influence on brain circuits maturation and the resulting aberrant behavior. Gestational factors (i.e., maternal 5-HT-ergic genotype, 5-HT-related diets, stress exposure, 5-HT-related medication intake, and immune activation) decrease (arrow blue, left) or increase (arrow red, right) the offspring’s brain 5-HT levels. The gestational factors in the middle (purple) have been found to increase as well as decrease brain 5-HT levels in the offspring. Gestational-factor-induced alterations in the formation and function of the prefrontal-limbic circuitry, thalamocortical circuitry, and the HPA-axis are depicted via orange, blue, and green dots, respectively. We mention which gestational factor has been associated to the highlighted neuropsychiatric disorders of ASD, ADHD, depression, and anxiety. Furthermore, where possible, direct links have been depicted between gestational-factor-induced 5-HT-ergic-related changes within one of the three brain circuits and a neuropsychiatric disorder. 5-HT: serotonin; ASD: autism spectrum disorders; ADHD: attention deficit hyperactivity disorder; HPA: hypothalamo-pituitary-adrenocortical; SSRIs: selective serotonin reuptake inhibitor.

**Table 1 ijms-21-05850-t001:** 5-HT receptor involvement in neurodevelopmental events.

Neurodevelopmental Events	5-HT Receptors Involved
Cell proliferation and survival/apoptosis	Postsynaptic 5-HT_1A/B/D & 2A & 6_ Astrocytic 5-HT_1-7_ Microglial 5-HT_2B/7_
Neuronal migration/positioning	Postsynaptic 5-HT_1A/B/D & 3 & 6_ Microglial 5-HT_2B/7_
Neuronal activity	Postsynaptic 5-HT_1A & 2A/C_ Astrocytic 5-HT_1-7_
Neuronal outgrowth/dendrite formation	Postsynaptic 5-HT_1A & 2A & 3 & 4 & 6 & 7_
Neuronal connectivity	Presynaptic 5-HT_2A_ Oligodendrocyte 5-HT_2A_
Synaptic formation/function	Astrocytic 5-HT_1-7_
Synaptic clustering	Presynaptic 5-HT_1B_ Postsynaptic 5-HT_1D_
Synaptic patterning	Microglial 5-HT_2B & 7_
5-HT neurotransmission (i.e., 5-HT clearance and release)	Presynaptic 5-HT_1A/B & 5A_ Microglial 5-HT_2B & 7_
HPA-axis activation	Postsynaptic 5-HT_1A & 2A/2C_

5-HT: serotonin; HPA: hypothalamo-pituitary-adrenocortical.

**Table 2 ijms-21-05850-t002:** Literature overview of associations between human maternal SSRI intake and child alterations. In depth information about the key characteristics of each article can be found in Appendix A. We included meta-analyses but also case-control and cohort studies that were not included in one of the meta-analyses.

Increased Risk of	Association Found	Association Not Found or Mainly Caused by the Underlying Maternal Psychiatric Condition
**ADHD**	-	Man et al. [189] (*meta-analysis*) Sujan et al. [190] Uguz [191] (*systematic review*)
**Anxiety/Depression**	Hermansen et al. [192] Lupattelli et al. [187] Malm et al. [188]	-
**ASD**	Andalib et al. [193] (*meta-analysis*) Brown et al. [194] (*meta-analysis*) Fatima et al. [195] (*review*) Gentile [196] (*review*) Kaplan et al. [197] (*meta-analysis*) Man et al. [198] (*meta-analysis*) Mezzacappa et al. [199] (*meta-analysis*)	Brown et al. [200] Kaplan et al. [201] (*meta-analysis*) Kobayashi et al. [202] (*meta-analysis*) Sujan et al. [190]

ADHD: attention deficit hyperactivity disorder; ASD: autism spectrum disorders.

**Table 3 ijms-21-05850-t003:** Literature overview of associations between perinatal SSRI exposure in rodents and behavioral changes in the offspring. In depth information about the key characteristics of each article can be found in Appendix A. Specifications are added when the SSRI effects were studied in gestational stressed animals.

Abnormalities	Association Found	Association Not Found
**Hyperactivity & impulsivity**	-	Khatri et al. [205] Ko et al. [206] Lisboa et al. [207] McAllister et al. [208] Olivier et al. [209]
**Increased depressive-like behavior**	Lisboa et al. [207] Ko et al. [206] Popa et al. [210] Rayen et al. [162] (*normalised stress effect*) Rebello et al. [7] Sprowles et al. [211] Zohar et al. [163] (*irrespective of stress effect*)	Altieri et al. [212] Gobinath et al. [213] McAllister et al. [208] Olivier et al. [209] Salari et al. [161] (*normalised stress effect*)
**Increased anxiety-like behavior**	Altieri et al. [212] Ansorge et al. [214] Ansorge et al. [215] Gobinath et al. [213] Khatri et al. [205] Olivier et al. [209] Rebello et al. [7] Smit-Rigter et al. [216] Sprowles et al. [211] Zohar et al. [163] (*irrespective of stress effect*)	Altieri et al. [212] Bairy et al. [217] Ehrlich et al. [218] Harris et al. [219] Kiryanova et al. [165] (*irrespective of stress effect*) Kiryanova & Dyck [220] Ko et al. [206] Lisboa et al. [207] McAllister et al. [208] Meyer et al. [221] Popa et al. [210] Salari et al. [161] (*normalised stress effect*) Silva et al. [222] Yu et al. [223] Zimmerberg & Germeyan [224]
**Repetitive behavior and sensory abnormalities**	Lee [225] Ko et al. [206] Maloney et al. [226] Sprowles et al. [211]	McAllister et al. [208]
**Decreased social behavior**	Bond et al. [227] Maloney et al. [226] (*social preference*) Ehrlich et al. [218] Khatri et al. [205] Olivier et al. [209] Rodriguez-Porcel et al. [228] Simpson et al. [229] Silva et al. [222] Yu et al. [223] Zimmerberg & Germeyan [224]	Houwing et al. [230] Gemmel et al. [150] (*normalised stress effect*) Gemmel et al. [160] (*irrespective of stress effect*) Ko et al. [206] Maloney et al. [226] (*social interaction*) Meyer et al. [221] Svirsky et al. [231] Zaidan et al. [232]

**Table 4 ijms-21-05850-t004:** Gestational factors influence multiple 5-HT-ergic systems. When the association between the gestational factor and the 5-HT system is proven by scientific studies, we report ‘YES’. When the association is theorized but not yet proven, we report ‘?’. When the association is not investigated, we report ‘Not Determined’ (N.D.).

	Placenta-Derived 5-HT-Ergic Systems			Offspring’s 5-HT-Ergic Systems	
	**Maternal Plasma Tryptophan and 5-HT Levels**	**Placental 5-HT Synthesis (i.e., Changes in Placental 5-HT Levels)**	**Maternal-Fetal Transport of Tryptophan and 5-HT**	**Brain 5-HT Production and Survival**	**5-HT Receptor or 5-HTT Activity/Gene Expression**
**Maternal 5-HT-ergic genotype**	YES	YES	?	YES	YES
**Maternal diet composition**	YES	YES	?	YES	YES
**Maternal stress**	YES	?	N.D.	YES	YES
**Maternal 5-HT-ergic medication intake**	YES	N.D.	?	N.D.	YES
**Maternal immune activation**	N.D.	YES	?	YES	YES

5-HT: serotonin; 5-HTT: serotonin transporter.

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
