# Peer review of "Gestational Factors throughout Fetal Neurodevelopment: The Serotonin Link"

_ijms, 2020, doi:10.3390/ijms21165850_

Round 1

Reviewer 1 Report

Authors have to be congratulate for such a huge work. The manuscript is well presented and of great interest, but some minor points are suggested in order to improve the overall level of this review.

A schematic representation of general information given in the first part of the 1.1 paragraph would be helpful, especially for students or young researchers (families of 5-HT receptors and usual signal transduction). In addition, a summary (in a table may be) of the pharmacological aspects of each 5-HT receptors and transduction could be of great interest, even if such information belong to academic knowledge: channel-receptor or metabotropic one?, activation or inhibition, calcium or tyrosine kinase or ... as transduction ...etc

Sometimes only correlations are mentioned which is of great interest and it is well presented, but the precise molecular mechanism with key actors (signaling pathway or gene expression or transcription factor (CREB?) or...) should be exposed (if known of course) in order for readers to establish a putative link with their own research. Example lines 343-344: "Such distortion in 5-HT-ergic neurons in the raphe nuclei could be prevented by maternal treatment with a 5-HT2A/2C receptor agonist or a 5-HT1A receptor agonist[119,125,128,129]" (especially true concerning ref 128, giving key molecular actors). An interesting summary is given in the discussion section and table 3 , but such mechanistic information would also be pertinent in the above sections.

The discussion section is sometimes only a summary of the scientific knowledge already exposed in the above sections. Since other sections are nicely presented, authors should propose more links between different studies in the same topic rather than descriptive summaries. On the contrary, the conclusion section present specific aspects that should be investigated in the light of convergent results obtained in different studies.

Reviewer 2 Report

The manuscript is a thorough review of the current literature regarding which and how gestational factors influence the establishment of the serotonin system in the developing, maturing and adult brain.

The authors include rodent and human studies to correlate abnormal embryonic and fetal serotonin network with neurodevelopmental cognitive disorders including ADHD and ASD.

The topic is extremely exciting and of high importance as the incidence of neurodevelopmental cognitive disorder increases. The analysis of the available literature has been done thoroughly and exhaustively. Whilst the effort is remarkable, the resulting manuscript appears more like a list of facts and fails to really synthesise the message the authors wish to convey. Basically, it's very dry.

My second major comment is that the information is often vague, with the authors referring to "changes" in certain networks without explaining the nature of the changes. Knowing that ASD is usually associated with hyperserotonemia and last into adulthood, while ADHD is associated with serotonin deficits and the symptoms resolve by adulthood, it is important to be more rigorous.

I recommend that the authors condense the information presented to make the manuscript more readable and the main points more salient. Also, the manuscript would benefit from placing the results described in the context of developmental/maturational processes taking place in the postnatal brain, in order to explain the changes observed. The perspective is very neurone centric when it is known that astrocytes and microglia are also sensitive to serotonin, and changes in this system has consequences on the establishment and remodelling of cortico-cortical and thalamoccortical networks.

Finally, the manuscript would benefit from including more illustrations.

Regarding the graphical abstract, it would be great to make more obvious that the boxed areas correspond to the gestational factors.

Minor comments:

Use "circuits" rather than "circuitries" throughout the manuscript

p.6 line 209 replace "earlier" by "previously"

p.7 line 300: The sentence starting with "Maternal..." needs to  be revised ("Neither" should be at the start but also the sentence is not clear)

All together, the authors have done a fantastic work researching the literature, tackling a very interesting topic.
